# Sero-Epidemiological Study of *Bordetella pertussis* Infection in the Italian General Population

**DOI:** 10.3390/vaccines10122130

**Published:** 2022-12-13

**Authors:** Tiziana Grassi, Francesco Bagordo, Marta Savio, Maria Cristina Rota, Francesco Vitale, Antonella Arghittu, Laura Sticchi, Giovanni Gabutti

**Affiliations:** 1Department of Biological and Environmental Sciences and Technologies, University of Salento, 73100 Lecce, Italy; 2Department of Pharmacy-Pharmaceutical Sciences, University of Bari, 70121 Bari, Italy; 3Post-Graduate School of Hygiene and Preventive Medicine, University of Ferrara, 44121 Ferrara, Italy; 4Department of Infectious Diseases, Italian Institute of Health (ISS), 00161 Roma, Italy; 5Department of Health Promotion, Mother and Child Care, Internal Medicine and Medical Specialties “G. D’Alessandro”, University of Palermo, 90127 Palermo, Italy; 6Department of Medicine, Surgery and Pharmacy, University of Sassari, 07100 Sassari, Italy; 7Department of Health Sciences, University of Genova, 16132 Genova, Italy; 8National Coordinator of the Working Group “Vaccines and Immunization Policies”, Italian Society of Hygiene, Preventive Medicine and Public Health, 16030 Cogorno, Italy

**Keywords:** sero-epidemiology, pertussis, general population, Italy

## Abstract

A multicenter study was conducted to estimate the prevalence of pertussis IgG antibodies (anti-PTx) in the Italian population. Serum samples (4154) collected in the years 2019–2020 from subjects aged 6 to 90 years were tested. The anti-PTx IgG levels were determined by ELISA test. The limit of detection was 5 IU/mL (International Units per milliliter); values ≥ 40 IU/mL and ≥100 IU/mL indicate an infection that has occurred in recent years and a recent infection (occurred during the last year), respectively. The mean concentration of anti-PTx IgG antibodies in the tested samples was 13 IU/mL; 1.0% of subjects had a titer ≥ 100 IU/mL, 5.3% a titer between 40 and 100 IU/mL, and 38.9% a titer < 5 IU/mL. The mean antibody concentration was significantly higher in males than in females. The age group 25–39 years had the lowest percentage of negative subjects (36.9%) and the highest prevalence of subjects with antibody titers ≥ 100 IU/mL (1.3%). In the age group ≥ 65 years, the prevalence of subjects with titers between 40 and 100 IU/mL (6.7%) and the percentage of negative subjects (44.8%) was higher than in the other age groups. The results highlight the possible role of adolescents and adults in the transmission of *B. pertussis*.

## 1. Introduction

The epidemiology of pertussis dramatically changed over the past century following the introduction of vaccination. In particular, the achievement of high vaccination coverage rates led to a significant decrease in the incidence and mortality related to the disease. However, the immunological pressure exerted by vaccination and the limited duration of immunity gave rise, especially in industrialized countries, besides a significant reduction in the incidence, to a different distribution of cases of the disease by age group and an important role for adolescents and adults in the transmission dynamics at the same time [1].

The fact that adolescents and adults can often represent an unrecognized source of infection requires the prevention of pertussis in infants, who are at the greatest risk of having severe and/or complicated, and potentially fatal, clinical forms [2]. For these reasons, effective pertussis prevention can only be achieved through the integration of different strategies.

Even if differences in available resources, vaccination policies and vaccination coverage among countries, along with the absence of detailed disease surveillance in many low- and middle-income countries (LMICs) do exist, the Global Pertussis Initiative urges improved surveillance in LMICs and recommends toddlers, adolescents, healthcare and childcare workers receive booster doses, where sufficient resources are available [3].

Among these, the use of booster doses plays a fundamental role in pregnant women, as specifically indicated in the 2017–2019 Italian National Immunization Plan (NIP) still in force [4]. In addition to boosters that guarantee vaccinal continuity, epidemiological surveillance should be improved.

The collection of data on the prevalence of antibodies in serum within a population (seroepidemiology) represents a fundamental tool to assess the impact of interventions implemented in the past, as well as to evaluate the ongoing risk of infectious diseases. This type of surveillance is particularly useful for vaccine-preventable infectious diseases allowing us to assess the achieved results and to define eventual alternative immunization strategies. Through serological investigations, it is possible to identify groups of individuals susceptible to a given infectious disease, allowing targeted preventive interventions [5]. Sero-epidemiological studies contributed to the elimination of poliomyelitis, the control of measles, rubella and the refinement of vaccination strategies, for example, for diphtheria in high-income countries [6]. To provide sound data, the collection of samples must be sufficiently large and organized in such a way as to be representative of the study population; moreover, the used assay must be adequate, standardized and performed in a laboratory with high-quality standards. Finally, the obtained results must be evaluated with the appropriate statistical surveys [7].

The epidemiology of some infectious diseases in Italy, as in other industrialized countries, significantly changed in recent decades thanks to the introduction of active immunization and the achievement of high vaccination coverage rates [8].

In agreement with the 2017–2019 NIP and to the recent law (May 2017) on compulsory immunizations, pertussis vaccination includes three compulsory doses in the first year of life (2 + 1 schedule with the hexavalent vaccine) and a fourth dose (as DTaP-IPV) at 5–6 years of age. Subsequently, decennial boosters with dTap are recommended starting from adolescence and in pregnant women during each pregnancy [4]. In Italy, pertussis is a mandatory notifiable disease. In 2017, 964 cases were reported, with a notification rate of 1.6 cases per 100,000 inhabitants, similar to the previous year. These data are in stark contrast with data reported by central and northern European countries, as well as by some southern countries such as Spain, where the notification rate is in line with or higher than the European one (9.4/100,000). Therefore, Italian data could be severely affected by a great under-notification and under-estimation. In particular, as pertussis in adolescents and adults is often mild, without specific symptoms, the disease is greatly underdiagnosed in these age groups. The consequence is that the real epidemiological impact of pertussis is not being perceived [9].

Antibodies produced against various *B. pertussis* antigens are considered to play a key role in disease protection. Unlike other vaccine-preventable diseases, no antibody level against a single antigen or a combination of antigens is definitively related to clinical protection. Nevertheless, some clinical trials using acellular vaccines containing three or five antigens, have shown that subjects with high antibody levels against pertussis toxin (PTx), pertactin (PRN) and fimbriae (FIM) are less likely to develop the disease in a clinically evident form when exposed to the pathogen [10].

Besides, it has been assumed that high anti-PTx antibody titers are indicative of recent or ongoing infection and this hypothesis has been supported by the group that undertook the European seroepidemiological study on vaccine-preventable diseases (ESEN) [7]. More recent studies confirmed that high levels of anti-pertussis antibodies (anti-PTx IgG, in particular) are indicative of recent infection and that *B. pertussis* spreading has significantly increased in adolescents and adults [11].

This seroepidemiological research, conducted among the Italian general population aims to obtain a more precise estimate of the real circulation of *B. pertussis* and to achieve relevant data to be used to fine-tune already ongoing immunization programs and to accelerate the control/elimination of pertussis.

## 2. Materials and Methods

### 2.1. Study Design and Sample Collection

The study was designed as an in vitro, not interventional, multicenter study, promoted by the Italian Institute of Health (ISS). The objectives of the study have been to evaluate the prevalence of antibodies to pertussis toxin (anti-PTx IgG) in the study population by age group, to evaluate the distribution of average antibody titers (GMT) (humoral immunity) by age group, to evaluate the distribution of subjects susceptible to pertussis as well as of subjects with high (>100 IU/mL) anti-PTx IgG levels by age and to evaluate the above points stratifying data accordingly to the gender and the geographical area (North, Center and South Italy).

Anonymous unlinked samples of residual sera from routine laboratory testing were collected from subjects between 6 and 90 years of age without any immune-depressive condition or any acute infection or having not recently undergone a blood transfusion. Some demographic data on each subject, such as age, gender and geographical area of residence, were taken. Collected sera were stored at −20 °C.

The total number of sera to be collected was calculated taking into account that the study has been designed to evaluate the sero-prevalence against a number of vaccine-preventable infectious diseases. The sampling protocol for each regional center was in accordance with the estimates made during the seroepidemiological studies conducted within the European ESEN (European Sero-Epidemiological Network) project [12,13,14,15,16], in which samples were taken for the national seroprevalence studies conducted in 1996, 2003–2004 and 2013–2014. Nowadays, available samples have been collected (period June 2019–May 2020) from 13 regional centers, (Northern Italy: Autonomous Province of Bolzano, Emilia-Romagna, Liguria, Piedmont and Veneto; Central Italy: Tuscany, Marche; Southern Italy and Islands: Basilicata, Calabria, Campania, Apulia, Sardinia, Sicily) (Figure 1).

### 2.2. Measurement of Antibody Titers

All collected sera have been sent to the Laboratory of Hygiene of the Department of Biological and Environmental Sciences and Technologies, University of Salento, Lecce, Italy where they have been analyzed. To determine the anti-PTx IgG levels, the classical immuno-assay Serion ELISA (Institut Virion/Serion GmbH, Würzburg, Germany) was used. This assay permits determining antibodies to *B. pertussis* specific toxin (PTx), using the first WHO International Standard Pertussis Antiserum (Human) 1st IS NIBSC code: 06/140, available since 2009 and expressing qualitative (positive/negative) and quantitative results in terms of antibody activity as international units per milliliter (IU/mL). The lower limit of detection for anti-PTx IgG is 5 IU/mL. A cut-off of 100 IU/mL is considered to be an indicator of a recent infection (occurred in the last year), while levels greater than or equal to 40 IU/mL are considered to be an indicator of an infection that has occurred in recent years [7,11].

### 2.3. Statistical Analysis

The main statistical techniques that apply to observational studies (epidemiological statistics) have been used to analyze the data collected in the study. In particular, the evaluation of the prevalence of anti-PTx IgG in the study population by age group has been conducted using the chi-square test. The same test has been used to compare data stratified accordingly to geographical area (Northern, Central and Southern Italy) and region. In any case, the significance level has been set at 0.05. The analysis of the average antibody titers (GMT) by age group, gender and geographical area has been conducted using a one-way ANOVA test followed by Tukey–Kramer posthoc test. As an additional analysis, the association between the presence of anti-PTx IgG antibodies and variables such as gender, age and the geographical area has been assessed through a logistic regression model.

## 3. Results

Overall, 4154 serum samples collected in the years 2019 and 2020 from subjects aged between 6 and 90 years (2010 males and 2144 females) residing in 13 Italian regions were tested: 1582 (38.1%) came from regions of Northern Italy (Autonomous Province of Bolzano, Emilia-Romagna, Liguria, Piedmont and Veneto), 430 (10.4%) from Central regions (Marche and Tuscany) and 2142 (51.6%) from the South and Islands (Basilicata, Calabria, Campania, Apulia, Sardinia and Sicily).

In relation to age, 715 (17.2%), 1213 (29.2%), 1277 (30.7%), 545 (13.1%) and 404 (9.7%) subjects were aged between 6 and 12 years, 13 and 24 years, 25 and 39 years, 40 and 64 years, and equal or more than 65 years, respectively.

The mean concentration of anti-PTx IgG antibodies in the tested samples was 13.0 IU/mL (95% CI, 12.4–13.6). Overall, 1.0% of subjects had an anti-PTx IgG titer ≥ 100 IU/mL and 5.3% a titer between 40 and 100 IU ml, while in 38.9% of the samples the antibody titer was undetectable (<5 IU/mL). The mean antibody concentration was significantly higher (*p* = 0.001) in the male population (14.1 ± 22.2 IU/mL) than in the female population (11.9 ± 17.1 IU/mL). Among males, 1.1% of subjects had an antibody titer ≥ 100 IU/mL and 6.2% a titer between 40 and 100 IU/mL, while 0.8% and 4.5% of females showed a titer ≥ 100 IU/mL and between 40 and 100 IU/mL, respectively (Table 1).

In relation to the age of the participants, although the distribution of seroprevalence (%) by antibody range was substantially overlapping (ns, *p* > 0.05) in the different age groups (Table 1), the average concentration of anti-PTx IgG antibodies was significantly different among groups (*p* = 0.011, one-way ANOVA) with particular reference to subjects aged 25–39 years (14.6 ± 21.8 IU/mL) and 6–12 years (12.5 ± 24.3 IU/mL) (Figure 2). Furthermore, the 25–39 age group recorded the lowest percentage of negative subjects (36.9%) and the highest prevalence of subjects with antibody titers ≥ 100 IU/mL (1.3%). Instead, in the ≥ 65 age class, the prevalence of subjects with antibody titers 40–100 IU/mL (6.7%) and the percentage of negative subjects (44.8%) was higher than in the other age groups.

On average, the anti-PTx IgG antibody titer appeared different (*p* = 0.001, one-way ANOVA) in the different Italian areas with higher values in the North (14.5 ± 22.6 IU/mL) and lower in the South and the Islands (11.8 ± 19.8 IU/mL) (Figure 3). The relative distribution of the tested samples as a function of the level of antibodies by geographical area (Table 1) appeared different as well. Overall, the highest prevalence of subjects with recent infection (IgG anti-PTx ≥ 100 IU/mL) occurred in Central Italy (1.9%) while the greatest proportion of subjects with titers between 40 and 100 IU/mL was recorded in Northern Italy (7.3%). In Southern Italy, on the other hand, the lowest prevalence of subjects with anti-PTx IgG antibody titer ≥ 100 IU/mL (0.5%) was found.

At the regional level (Table 2) a different mean concentration of anti-PTx IgG antibodies (*p* < 0.05) and a different distribution of seroprevalence (*p* < 0.05) were highlighted. The highest mean antibody titers were recorded in Veneto (19.5 ± 37.6 IU/mL) and in the Autonomous Province of Bolzano (17.7 ± 22.4 IU/mL). The prevalence of subjects with titer ≥ 100 IU/mL was higher in Veneto (2.9%), while a higher proportion of subjects with a titer between 40 and 100 IU/mL was shown in the Autonomous Province of Bolzano (13.3%).

## 4. Discussion

It is well known that immunity, natural or acquired with vaccination, against *B. pertussis* is not lifelong and tends to decline over time; currently, there are no known antibody levels, against a single antigen or a combination of antigens, which can be correlated with certainty with clinical protection [17,18].

Globally, the WHO reports a worldwide vaccination coverage rate of 83% for the 3-dose cycle (with DTP) in 2020 and estimated over 150,000 pertussis cases in 2018 [19]. In Europe, 35,627 cases were reported in 2018.

Individuals aged ≥15 years accounted for 62% of all reported cases: infants < 1 year of age, too young to be fully vaccinated, were the most affected age group, with the highest incidence rate (44.4 per 100,000 inhabitants and three reported deaths), followed by children between the ages of 10 and 14. European data also confirm that the clinical presentation of pertussis in adolescents and adults can be mild and often not diagnosed, thus involving the risk of transmission for children, who are too young to have started or completed the primary course of vaccination [20].

In Italy, the epidemiological trend changed following the recommendation for the anti-pertussis vaccination by the Ministry of Health in 1962. Subsequently, the introduction of the acellular vaccine, in combination with the antigens for diphtheria and tetanus, facilitated a further reduction in notified cases, which moreover have increased since 2016. The comparison with other countries suggests that pertussis in Italy is considerably underestimated. Most cases occur during the first year of life and a second peak occurs in adolescence, an expression of the decline in immune protection. Adolescents and adults play an important role in the dynamics of infection, as also demonstrated by numerous seroepidemiological studies [11,21,22,23,24,25].

In 2015, Barkoff AM et al. published a paper on the seroprevalence of pertussis in different immunized populations in the world. Firstly, they pointed out the relevance of measuring anti-PTx IgG antibodies, with PTx being the only specific antigen of *B. pertussis*. Another relevant point was the significant underestimation of pertussis cases in all studied countries, notwithstanding they adopted whole or acellular pertussis vaccine [26].

A study conducted in 18 European countries showed that 2.7–5.8% of the samples analyzed (range 0–9.7%) and belonging to subjects aged 40–59 years had an anti-PTx IgG level > 100 IU/mL, indicative of recent exposure to *B. pertussis*. These data confirm the circulation of the microorganism in the evaluated age groups despite the achievement of high levels of vaccination coverage and can be considered indicative of a general underestimation of the case series as well [27].

Another recently published European study reported on pertussis seroprevalence among 20–39-year-old subjects in 14 countries. The rate of subjects with a titer indicative of recent infection (>100 IU/mL) ranged between 0.2 and 5.7%. Italy and Portugal showed high rates of subjects with a titer equal to 50–100 IU/mL (13.9 and 12.3%, respectively), which Authors considered indicative of an infection that occurred a few years before [28].

Underreporting of pertussis cases, changing epidemiology, circulation of *B. pertussis* in highly vaccinated populations, and the need for boosters have been pointed out in several other papers [29,30,31,32].

Similar considerations had been made in Italy. Indeed, Palazzo R et al. had highlighted an increased circulation of *B. pertussis* in the Italian adult population on the basis of the seroprevalence data collected in the period 2012–2013. While using a different method than that used in the present study, >80% of the adult population had detectable levels of anti-PTx IgG; in particular, 9.1% and 5% of adults had anti-PTx IgG titers equal to 50–99 IU/mL and >100 IU/mL, respectively [11]. Another study was performed in Italy to evaluate the epidemiological dynamics of pertussis in families; seroprevalence in parents allowed to clarify their role in the spreading of the pathogen [33].

The seroprevalence against *B. pertussis* has been evaluated in Tuscany in the periods 1992–2005 and 2013–2016 showing that since 2002 at least 50% of >22 years-old subjects had low IgG titers (<50 IU/mL) highlighting the risk of being susceptible to pertussis [25].

Data collected in the present work, therefore, fall within the ranges reported in the works previously described.

In the present study, IgG titers of males resulted higher than those of females but it is not possible to explain this difference between genders. In this respect, it should be pointed out that different studies have provided different and contradictory results regarding both seroprevalence and antibody titers in males and females [25,27,34,35,36,37].

Similarly, it is not possible to provide a scientifically sound interpretation of the differences in seroprevalence observed in the various Italian geographical areas and regions, taking into account that there are no indications of different geographical patterns relating to the circulation of *B. pertussis* in Italy nor in the application of the recommendations provided by the Italian national immunization plan [4].

In this context, booster vaccinations represent the operational response to the need to guarantee vaccination continuity, i.e., the long-term protection of subjects successfully vaccinated, but who in some cases may be exposed to new risks of contracting the infection and/or disease due to the decay of the immune protection provided by vaccination, the reduced possibility of natural boosters or the reintroduction of a pathogen from endemic areas [38,39].

Concerning underreporting, data from the present study seem to confirm what has been observed in other countries and/or in different settings and confirm the need for more effective epidemiological surveillance [40,41]. Relevant surveillance gaps have been reported from several LMIC countries as well [42].

As regards pertussis (tetanus and diphtheria), the recommendation to carry out boosters at ten-year intervals starting from adolescence allows maintaining the effectiveness of the vaccination previously received with the primary cycle of the pediatric age.

As a matter of fact, it should be borne in mind that the interruption of the transmission of the infection can only be pursued by achieving high levels of immunity in all age groups of the population. In addition to the objective of preventing the pathology in early childhood and the related clinically severe forms, attention must also be paid to controlling the transmission of the infection. In industrialized countries, in the face of the significant reduction in the incidence, a different distribution of cases by age group of the disease and an important role for adolescents and adults in the dynamics of transmission was recorded [43].

Pertussis continues to be a relevant health problem for the most vulnerable subjects, such as infants and children who are unvaccinated, or not fully immunized, and exposed to contact with adolescents and adults who act as an often-unidentified source of infection. Prevention of whooping cough requires an integrated approach. To increase and maintain high levels of vaccination coverage, different immunization strategies have been defined and/or already adopted in various European countries that provide vaccination for newborns, preschool children, adolescents and adults, health workers, childhood, pregnant women and cocoon strategy [33,43,44].

Finally, it should be remembered that vaccination boosters are indicated for everyone, including frail people and patients suffering from chronic degenerative diseases, who are in any case at an increased risk of post-infectious complications [45,46,47].

The need for an integrated approach to pertussis prevention is urgent also in LMIC countries, where the re-emergence of pertussis should be usually related to the lack of administrative capacity. From this perspective, as sustainable and equitable health improvements are the product of effective policy at all levels of government and collaborative efforts between all spheres of society, countries and regions should define common goals and investments between the health sector and other sectors in order to significantly improve health and well-being while respecting social and health equity [48]. Besides, the high infant mortality rate in LMICs supports maternal vaccination [49]. Noteworthy, in LMIC countries immunization delivery costs should be used for assessing the cost-effectiveness and strategic planning needs of immunization programs [50].

The design of this study presents some limitations. As it is based on a convenience sample, data obtained could not be representative of the Italian population as a whole; nevertheless, these data will give an indication of the rate of sero-positivity/-negativity according to age groups that can be expected in larger surveys. Secondly, no serological marker of protection has been established for pertussis. However, serological testing by means of ELISA tests allows for quantifying anti-PTx IgG. Some authors have proposed the determination of antibodies directed against PTx and provided recommendations for the interpretation of the measured antibody activities. Results greater than 100 IU/mL in adolescents and adults are indicative of recent contact with *B. pertussis*, while values below 40 IU/mL allow, with a good degree of certainty, to exclude infection [7,51].

Thirdly, having no data available on any previous vaccination, it was not possible to evaluate the probability of infection vs. the probability of vaccination by age and sex.

In conclusion, as results greater than 100 IU/mL of anti-PTx IgG in adolescents and adults are indicative of recent contact with *B. pertussis,* this would highlight the possible role of these subjects as a source of infection, supporting the use of boosters in adolescents, adults and pregnant women.

## Figures and Tables

**Figure 1 vaccines-10-02130-f001:**
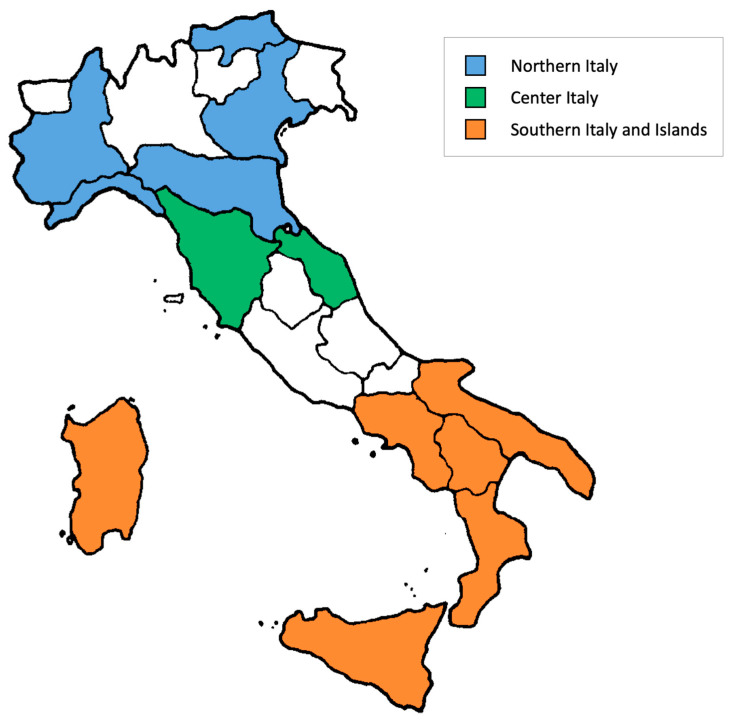
Regions participating in the study divided by geographical area.

**Figure 2 vaccines-10-02130-f002:**
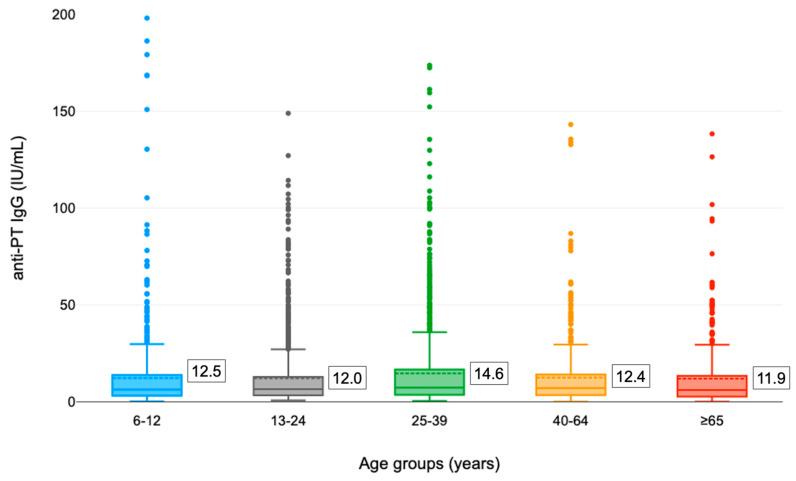
Distribution of anti-PTx IgG titers in different age groups (the mean concentration is shown in the box).

**Figure 3 vaccines-10-02130-f003:**
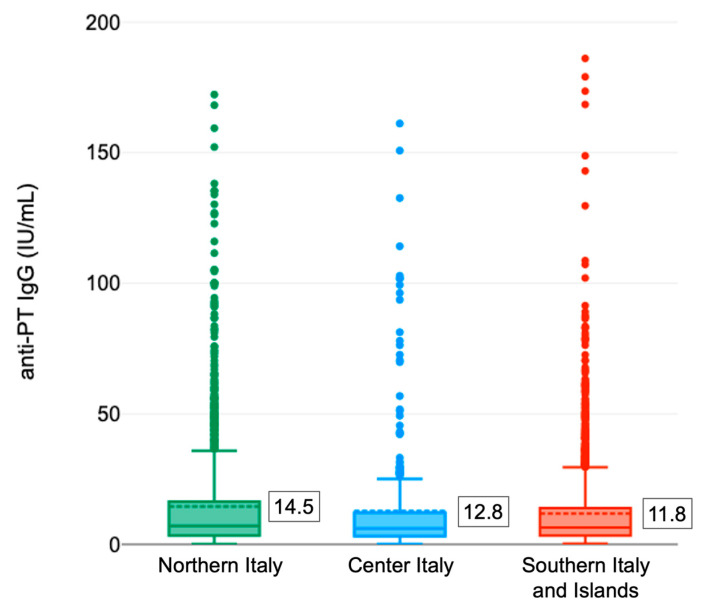
Distribution of anti-PTx IgG titers in different Italian areas (the mean concentration is shown in the box). Legenda: Northern Italy: Bolzano, Emilia Romagna, Liguria, Piedmont and Veneto; Center Italy: Marche and Tuscany; Southern Italy and Islands: Basilicata, Calabria, Campania, Apulia, Sardinia and Sicily.

**Table 1 vaccines-10-02130-t001:** Results of the serological investigations carried out on 4154 subjects stratified by gender, age group and geographical area.

	Samples N	Mean IgG Titer ± SD (IU/mL)	<5 IU/mL %	5–40 IU/mL %	40–100 IU/mL %	≥100 IU/mL %	*p*-Value *
**Gender**							
Males	2010	14.1 ± 22.2	36.8	55.8	6.2	1.1	0.007
Females	2144	11.9 ± 17.1	40.9	53.8	4.5	0.8	
**Age group (years)**							
6–12	715	12.5 ± 24.3	41.1	54.0	3.8	1.1	0.993
13–24	1213	12.0 ± 16.2	38.7	55.8	4.9	0.7	
25–39	1277	14.6 ± 21.8	36.9	55.6	6.2	1.3	
40–64	545	12.4 ± 17.0	37.1	56.7	5.5	0.7	
≥65	404	11.9 ± 17.1	44.8	47.8	6.7	0.7	
**Geographical area**							
Northern Italy	1582	14.5 ± 23.6	37.9	53.5	7.3	1.3	<0.001
Center Italy	430	12.8 ± 23.5	43.3	50.9	4.0	1.9	
Southern Italy and Islands	2142	11.8 ± 19.8	38.8	56.5	4.2	0.5	
**TOTAL**	**4154**	**13.0 ± 19.8**	**38.9**	**54.8**	**5.3**	**1.0**	

* by chi-square test.

**Table 2 vaccines-10-02130-t002:** Mean antibody titer and relative distribution (%) of the tested samples as a function of the level of anti-PTx IgG antibodies by region.

Region	Samples N	Mean IgG Titer ± SD IU/mL	<5 IU/mL %	5–40 IU/mL %	40–100 IU/mL %	≥100 IU/mL %
Apulia	362	11.5 ± 17.1	39.2	56.9	3.0	0.6
Basilicata	308	10.5 ± 13.7	44.2	51.3	4.2	0.3
Bolzano	353	17.7 ± 22.4	33.4	52.1	13.3	1.1
Calabria	363	12.5 ± 17.8	40.2	53.7	5.8	0.3
Campania	275	9.1 ± 10.3	47.3	50.2	2.5	0.0
Emilia-Romagna	364	13.4 ± 20.4	37.1	56.3	5.2	1.4
Liguria	328	12.1 ± 15.3	40.2	53.4	6.1	0.3
Marche	99	9.2 ± 17.5	53.5	43.4	1.0	2.0
Piedmont	363	12.4 ± 20.0	41.6	53.7	3.0	1.7
Sardinia	396	15.3 ± 21.1	27.8	66.9	3.8	1.5
Sicily	438	11.1 ± 14.5	38.4	56.6	4.8	0.2
Tuscany	331	13.9 ± 24.9	40.2	53.2	4.8	1.8
Veneto	174	19.5 ± 37.6	36.2	50.0	10.9	2.9

## Data Availability

The data supporting the findings of this study are contained within the article.

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
