# Peer review of "Sero-Epidemiological Study of Bordetella pertussis Infection in the Italian General Population"

_vaccines, 2022, doi:10.3390/vaccines10122130_

Round 1
Reviewer 1 Report
I have carefully read the article entitled "Sero-epidemiological study of Bordetella pertussis infection in the Italian general population" by Grassi T el al.
I found it to be a simple but very interesting and necessary article in these times of rebirth of some predictable and curable infectious diseases.
The idea and the objective of studying the prevalence of antibodies to pertussis toxin (anti-PTx IgG) in the study population by age group, have seemed clear to me. The introduction has been well written with adjusted referennes. The established method has been well explained and developed. The results are shown effectively and with correct tables and figures
The discussion has been extensive but interesting and has also considered the possible limitations of the study together with the aspects to consider in the future as the authors reflect in the final conclusions.
Author Response
Reviewer 1
I have carefully read the article entitled "Sero-epidemiological study of Bordetella pertussis infection in the Italian general population" by Grassi T el al.
I found it to be a simple but very interesting and necessary article in these times of rebirth of some predictable and curable infectious diseases.
The idea and the objective of studying the prevalence of antibodies to pertussis toxin (anti-PTx IgG) in the study population by age group, have seemed clear to me. The introduction has been well written with adjusted referennes. The established method has been well explained and developed. The results are shown effectively and with correct tables and figures
The discussion has been extensive but interesting and has also considered the possible limitations of the study together with the aspects to consider in the future as the authors reflect in the final conclusions.
Thank you for this good evaluation of our manuscript.
Reviewer 2 Report
A simple study nothing extraordinary. Here the authors have carried out a seroepidemiological study focusing Italian general population on Bordetella pertussis infection. I have a few comments as follows:
Please show a sample size calculation to justify that 363 and 4,154 sample is enough for this type of study.
Some of the demographic data is missing for example ethinicity/race...can it be added??
In addition, it will be good if the probability of infection vs. the probability of vaccination by age and sex can be shown.
IMPORTANTLY,\ the discussion section is too general and needs to be rewritten more specifically focusing the actual findings of the study.
Author Response
Reviewer 2
A simple study nothing extraordinary. Here the authors have carried out a seroepidemiological study focusing Italian general population on Bordetella pertussis infection. I have a few comments as follows:
Please show a sample size calculation to justify that 363 and 4,154 sample is enough for this type of study.
We have already included this kind of information in the Materials and Methods section (page 3, lines 122-128) as follows: “The total number of sera to be collected was calculated taking into account that the study has been designed to evaluate the seroprevalence against a number of vaccine-preventable infectious diseases. The sampling protocol for each regional center was in accordance with the estimates made during the serum-epidemiological studies conducted within the European ESEN (European Sero-Epidemiological Network) project [12-16], in which samples were taken for the national seroprevalence studies conducted in 1996, 2003-2004 and 2013- 2014.”.
Please, take note that the same sampling methodology has been applied during all the sero-epidemiological studies performed in 1996, 2003-2004 and 2013-2014 and subsequently published. In order to address this point, we have added some references that would help to clarify.
The references added (n. 12-16) are the following (page 3, line 126):
- The development of a European sero-epidemiology collaboration for the investigation of vaccine preventable disease: the European Sero-Epidemiology Network. Biomed 1996–1999. Contract number: PL95- 1039.
- Edmunds WJ, Pebody RG, Aggerback H, et al. The sero-epidemiology of diphtheria in Western Europe. ESEN Project. European Sero-Epidemiology Network. Epidemiol Infect. 2000; 125:113-25.
- de Melker H, Pebody RG, Edmunds WJ, et al. The seroepidemiology of measles in Western Europe. Epidemiol Infect. 2001; 126:249-59.
- Pebody RG, Gay NJ, Giammanco A, et al. The seroepidemiology of Bordetella pertussis infection in Western Europe. Epidemiol Infect. 2005; 133:159-71.
Of course, References numbering in the manuscript has been completely updated.
Some of the demographic data is missing for example ethnicity/race...can it be added??
In the Materials and Methods we have specified that “Some demographic data of each subject, such as age, gender and geographical area of residence, were taken.” (Page 3, lines 119-121).
We are sorry, but we do not have any data about ethnicity/race or else.
In addition, it will be good if the probability of infection vs. the probability of vaccination by age and sex can be shown.
It was not possible to have data on vaccinal status of evaluated subjects and for this reason it is not possible to address this point. We have added the following sentence in the discussion section: “Thirdly, having no data available on any previous vaccination, it was not possible to evaluate the probability of infection vs. the probability of vaccination by age and sex.” (Page 12, lines 383-384).
IMPORTANTLY, the discussion section is too general and needs to be rewritten more specifically focusing the actual findings of the study.
Taking into account this point, as well as those made by other Reviewers, we have changed some parts of the Discussion section. Please see pages 11 and 12.
Reviewer 3 Report
Dear Authors,
Submission entitled:
"Sero-epidemiological study of Bordetella pertussis infection in the Italian general population "
is indeed a valuable one tackling these issues in a large Mediterranean nation.
It may fill certain knowledge gaps in the seminal literature in this area.
Yet it has one core bottleneck weakness.
Its evidence base is overtly homogeneous too heavily relying exclusively on sources originating fro mwealthy OECD countries.
Yet by far the most of global burden of Bordetella Pertusis and other major communicable infectious diseases remains with the LMICs nations of the Global South.
Thus introduction and discussion should be expanded to encircle far more evidence related to the subject of research coming from major LMICs countries and in particular Emerging BRICS Markets as leading ones among them.
For this purpose I warmly recommend consideration for inclusion of at least several pieces proposed below alongside few additional ones at authors own disposal:
https://www.ncbi.nlm.nih.gov/pmc/articles/PMC8637761/
https://www.ncbi.nlm.nih.gov/pmc/articles/PMC7585857/
https://www.tandfonline.com/doi/abs/10.1080/14737167.2022.2086122
https://globalizationandhealth.biomedcentral.com/articles/10.1186/s12992-020-00590-3
https://link.springer.com/article/10.1186/s12961-022-00822-5
https://www.mdpi.com/2071-1050/13/19/11038
https://www.tandfonline.com/doi/full/10.1080/13696998.2019.1600523
https://www.mdpi.com/1660-4601/16/17/3043
https://www.mdpi.com/2071-1050/13/6/3415
https://www.tandfonline.com/doi/full/10.1080/13696998.2021.2007691
https://www.mdpi.com/1660-4601/17/24/9404/htm
https://www.tandfonline.com/doi/full/10.1080/13696998.2021.2013675
https://www.frontiersin.org/articles/10.3389/fpubh.2022.970922/full
https://www.tandfonline.com/doi/full/10.1080/13696998.2021.2007691
https://www.tandfonline.com/doi/full/10.1080/13696998.2022.2054202
PS. I remain willing to review the revised manuscript.
Author Response
Reviewer 3
Submission entitled:
"Sero-epidemiological study of Bordetella pertussis infection in the Italian general population " is indeed a valuable one tackling these issues in a large Mediterranean nation.
It may fill certain knowledge gaps in the seminal literature in this area.
Yet it has one core bottleneck weakness.
Its evidence base is overtly homogeneous too heavily relying exclusively on sources originating fro mwealthy OECD countries.
Yet by far the most of global burden of Bordetella Pertusis and other major communicable infectious diseases remains with the LMICs nations of the Global South.
Thus introduction and discussion should be expanded to encircle far more evidence related to the subject of research coming from major LMICs countries and in particular Emerging BRICS Markets as leading ones among them.
For this purpose I warmly recommend consideration for inclusion of at least several pieces proposed below alongside few additional ones at authors own disposal:
https://www.ncbi.nlm.nih.gov/pmc/articles/PMC8637761/
https://www.ncbi.nlm.nih.gov/pmc/articles/PMC7585857/
https://www.tandfonline.com/doi/abs/10.1080/14737167.2022.2086122
https://globalizationandhealth.biomedcentral.com/articles/10.1186/s12992-020-00590-3
https://link.springer.com/article/10.1186/s12961-022-00822-5
https://www.mdpi.com/2071-1050/13/19/11038
https://www.tandfonline.com/doi/full/10.1080/13696998.2019.1600523
https://www.mdpi.com/1660-4601/16/17/3043
https://www.mdpi.com/2071-1050/13/6/3415
https://www.tandfonline.com/doi/full/10.1080/13696998.2021.2007691
https://www.mdpi.com/1660-4601/17/24/9404/htm
https://www.tandfonline.com/doi/full/10.1080/13696998.2021.2013675
https://www.frontiersin.org/articles/10.3389/fpubh.2022.970922/full
https://www.tandfonline.com/doi/full/10.1080/13696998.2021.2007691
https://www.tandfonline.com/doi/full/10.1080/13696998.2022.2054202
Thank you for the evaluation of our manuscript.
Taking into account the request to include some evidences coming from LMIC countries, we have added some parts both in the Introduction and in the Discussion section. Moreover, we have used one of the proposed references, since the others were mainly related to economical evaluations and we inserted some different references as suggested in order to better describe the framework.
In detail, in the Introduction section we have added the following part:
“Even if differences in available resources, vaccination policies and vaccination among countries, along with the absence of detailed disease surveillance in many low- and middle-income countries (LMICs) do exist, the Global Pertussis Initiative urges improved surveillance in LMICs and recommends for toddlers, adolescents, healthcare and childcare workers booster doses, where sufficient resources are available [3].” (Pages 1 and 2, lines 44-48).
In the Discussion section, we have added the following part:
- “Relevant surveillance gaps have been reported from several LMIC countries as well [42].” (Page 11, lines 339-340).
- “The need for an integrated approach to pertussis prevention is urgent also in LMIC countries, where the re-emergence of pertussis should be usually related to the lack of administrative capacity. In this perspective, as sustainable and equitable health improvements are the product of effective policy at all levels of government and collaborative efforts between all spheres of society, countries and regions should define common goals and investments between the health sector and other sectors in order to significantly improve health and well-being while respecting social and health equity [48]. Besides, the high infant mortality rate in LMICs supports maternal vaccination [49]. Noteworthy, in LMIC countries immunization delivery costs should be used for assessing the cost-effectiveness and strategic planning needs of immunization programs [50].” (Pages 11 and 12, lines 363-372).
Reviewer 4 Report
In this manuscript, the authors investigated anti-PT IgG seroprevalence among Italian population during 2019 to 2020 (n = 4,719). It was written that the age group 25-39 years was found to have highest antibody titers compared to other age groups, indicating asymptomatic infection. Therefore, the authors concluded that adolescent and adults may be a source of B. pertussis transmission to immunologically vulnerable people. Considering the authors’ logic, Figure 2 (seroprevalence of anti-PT IgG by age group) is the most important data in this study. However, titer distributions appear to be rarely different from each age group, and statistical technique used here had some concerns. I feel that this manuscript is constructed conclusion first, and results interpretation is adjusted in complying with previous studies.
Specific remarks:
Line numbers should be displayed.
Page 1, Abstract: Sample number involved in this study (n = 4,154?) is better to be included in the abstract to indicate an investigation scale.
Page 3, line 1, Materials and Method: I think currently there is no reliable ELISA kit targeting anti-B. parapertussis IgG.
Page 3, Table 1: I did not understand the importance of this table. 13 regional centers × 363 samples = 4,719 samples, but actually 4,154 serum samples were only investigated in this study. Instead, I recommend showing sample numbers in each age group at each regional center. Further, age category should be conformed to that of Figure 2.
Page 4, lines 10-19: The authors used commercially available ELISA kit, and therefore this paragraph is not necessary to be written.
Page 5, Results (the bottom paragraph) and page 6, Figure 2: The authors did not find any significant difference overall by one-way ANOVA (P > 0.05). However, subsequently they found a significant difference between 6-12 years and 25-39 years groups (P = 0.011)?? Which post-hoc test did the authors perform? Multiple comparison problem is concerned if t-test was repeatedly used.
Page 6, Figure 2: Age group 6-12 years?
Page 7, Figure 3: I did not understand the significance of this figure. The data are likely to overlap with Table 3.
Page 7, Table 3: Mean IgG titer ± SD (IU/ml)
Page 8-10, Discussion: Please provide clearly your interpretation of your findings first. Then, compare your findings to previous key studies, and highlight the significance and/or mention limitation of this study. Basically, I recommend discussing your findings by citing references; why IgG titers of males are higher than those of females? Why 25-39 years group had higher titers than other age groups (if it is statistically correct), why people in northern Italy had highest titers? what do you think about the data discrepancy between Wehlin L et al (Ref-23) and this study; 13.9% of 20-39 years group had 50-100 IU/ml in Ref-23, whereas only 6.2% of 25-39 years group had 40-100 IU/ml in this study.
Further, what do you think about these impacts on public health?
Author Response
Reviewer 4
In this manuscript, the authors investigated anti-PT IgG seroprevalence among Italian population during 2019 to 2020 (n = 4,719). It was written that the age group 25-39 years was found to have highest antibody titers compared to other age groups, indicating asymptomatic infection. Therefore, the authors concluded that adolescent and adults may be a source of B. pertussis transmission to immunologically vulnerable people. Considering the authors’ logic, Figure 2 (seroprevalence of anti-PT IgG by age group) is the most important data in this study. However, titer distributions appear to be rarely different from each age group, and statistical technique used here had some concerns. I feel that this manuscript is constructed conclusion first, and results interpretation is adjusted in complying with previous studies.
Thank you for the evaluation of our manuscript. We totally disagree that “the manuscript is constructed conclusion first”. The points regarding titer distributions and statistical technique have been addressed (please see answers to specific remarks).
Specific remarks:
Line numbers should be displayed.
Line numbers have been added.
Page 1, Abstract: Sample number involved in this study (n = 4,154?) is better to be included in the abstract to indicate an investigation scale.
Sample number has been added (page 1, line 18).
Page 3, line 1, Materials and Method: I think currently there is no reliable ELISA kit targeting anti-B. parapertussis IgG.
This sentence has been changed as follows (page 4, lines 155-157):
“No serological correlates of protection have yet been established for pertussis; however, available ELISA assays allow to determine human IgG antibodies directed against B. pertussis in serum and plasma.”
Page 3, Table 1: I did not understand the importance of this table. 13 regional centers × 363 samples = 4,719 samples, but actually 4,154 serum samples were only investigated in this study. Instead, I recommend showing sample numbers in each age group at each regional center. Further, age category should be conformed to that of Figure 2.
Table 1 was included in order to explain the sampling size used. Taking into account the point raised by the Reviewer, we have deleted Table 1 and included sample numbers at each regional level in Table 2 (previously Table 3, page 9).
Besides, the title of figure 2 (page 7) has been modified and now reports the correct age range (6-12 years).
Page 4, lines 10-19: The authors used commercially available ELISA kit, and therefore this paragraph is not necessary to be written.
We have changed the placement of some sentences in order to make the text clearer.
In detail:
“The objectives of the study have been to evaluate the prevalence of antibodies to pertussis toxin (anti-PTx IgG) in the study population by age group, to evaluate the distribution of average antibody titers (GMT) (humoral immunity) by age group, to evaluate the distribution of subjects susceptible to pertussis as well as of subjects with high (>100 IU/ml) anti-PTx IgG levels by age and to evaluate the above points stratifying data accordingly to the gender and the geographical area (North, Center and South Italy).” has been included in page 3 (lines 111-116).
“No serological correlates of protection have yet been established for pertussis; however, available ELISA assays allow to determine human IgG antibodies directed against B. pertussis in serum and plasma. These assays permit to determine antibodies to B. pertussis specific toxin (PTx), using the first WHO International Standard Pertussis Antiserum (Human) 1st IS NIBSC code: 06/140, available since 2009 and expressing qualitative (positive/negative) and quantitative results in terms of antibody activity as international units per milliliter (IU/ml).” has been included in page 4 (lines 155-161).
The following part has been deleted:
“The test strips of the ELISA microtiter plate are coated with specific antigens of the pathogen of interest (PT antigen). If the patient's serum contains specific anti-PT IgG antibodies, these bind to the antigen fixed on the strips. A secondary antibody, conjugated with alkaline phosphatase enzyme, detects and binds the immune complex. The signal intensity of the reaction product is proportional to the concentration of the analyte in the sample and is measured with a photometric method.”
Page 5, Results (the bottom paragraph) and page 6, Figure 2: The authors did not find any significant difference overall by one-way ANOVA (P > 0.05). However, subsequently they found a significant difference between 6-12 years and 25-39 years groups (P = 0.011)?? Which post-hoc test did the authors perform? Multiple comparison problem is concerned if t-test was repeatedly used.
We did not find any significant difference (p>0.05) in the distribution of seroprevalence (%) by antibody range (<5, 5-40, 40-100, >100 IU/mL) in the different age groups by using the chi-square test. On the contrary, we found a significant different mean antibody concentration among the different age groups (p=0.011) when we used one-way ANOVA. The Tukey-Kramer post-hoc test was subsequently performed to make the multiple comparison. To better clarify this result, some specifications have been added to the test (page 6, lines 216 and 218) and the information about the post-hoc test have been included in the Materials and Methods section (page 5 line 188).
Page 6, Figure 2: Age group 6-12 years?
The title of Figure 2 (page 7) has been modified and now reports the correct age range (6-12 years).
Page 7, Figure 3: I did not understand the significance of this figure. The data are likely to overlap with Table 3.
In Table 2 (previously table 3, page 9), data relating to the geographical areas have been eliminated. In this way in Table 2 are only shown the regional data while data relating to the geographic areas are included in Figure 3 (page 9). Besides, the regions included in each geographical area have been added in the caption of Figure 3.
Page 7, Table 3: Mean IgG titer ± SD (IU/ml)
Thank you. “Mean IgG titer ± SD (UI/ml)” has been changed in “Mean IgG titer ± SD (IU/ml)”
Page 8-10, Discussion: Please provide clearly your interpretation of your findings first. Then, compare your findings to previous key studies, and highlight the significance and/or mention limitation of this study. Basically, I recommend discussing your findings by citing references; why IgG titers of males are higher than those of females? Why 25-39 years group had higher titers than other age groups (if it is statistically correct), why people in northern Italy had highest titers? what do you think about the data discrepancy between Wehlin L et al (Ref-23) and this study; 13.9% of 20-39 years group had 50-100 IU/ml in Ref-23, whereas only 6.2% of 25-39 years group had 40-100 IU/ml in this study.
Some changes have already been included in the Discussion section taking into account the points raised also by another Reviewer (page 11, lines 339-340; pages 11 and 12, lines 363-372). Besides, some findings have been discussed in more detail in order to clarify the results of our study.
We would not like to change the structure of the Discussion section.
In detail, the following parts have been included as well as new references:
“In the present study, IgG titers of males resulted higher than those of females but it is not possible to explain this difference between genders. In this respect, it should be pointed out that different studies have provided different and contradictory results both as regards seroprevalence and antibody titers in males and females [25,27,34-37].
Similarly, it is not possible to provide a scientifically sound interpretation of the differences in seroprevalence observed in the various Italian geographical areas, taking into account that there are no indications of different geographical patterns relating to the circulation of B. pertussis in Italy nor in the application of the recommendations provided by the Italian national immunization plan [4].” (Page 11, lines 322-330).
Taking into account the point related to the discrepancy between data by Wehlin L. et al (28) and this study related to subjects of 20-39 years group with a titer equal to 50-100 IU/ml (13.9% vs 6.2%), this latter could be related to the different sample size analyzed (about 250 subjects in the study by Wehlin collected in 5 Italian cities vs. more than 2400 samples in the age groups 13-24 and 25-39 years of age collected in 13 Regions in the present research) and to the different method used.
Noteworthy the Italian data included in Whelin’s paper has been separately published in a different paper already included in our references as well as in the Discussion section (Palazzo R, Carollo M, Fedele G et al. Evidence of increased circulation of Bordetella pertussis in the Italian adult population form seroprevalence data (2012-2013). JMM 2016; 65:649-57). (page 10-11, lines 309-321)
For these reasons we would not add more nor change this part.
Further, what do you think about these impacts on public health?
If this point has been correctly understood, we believe that the impact and the implications of these data on public health have been addressed. We have pointed out that there is the urgent need to implement the prevention of pertussis adopting an integrated immunization strategy.
Round 2
Reviewer 4 Report
This study includes large-scale sero-epidemiology data, and the data it-self is valuable for public health. However, the manuscript is still required extensive improvements mainly in data presentation.
Specific remarks:
Lines 102-163, Materials and Methods: Some descriptions are duplicated (e.g., lines 85-87 and line134, lines 134-140 and 144-149, line 114 and 133). Please eliminate such duplicating expressions throughout the manuscript. To avoid such errors, I recommend describing methodological details under each block: e.g., 1. Study design and sample collection, 2. Measurement of antibody titers, 3. Statistical analysis.
Lines 149-152: Please provide the reference for this titer categorizing.
Figure 1: Explanation of color keys should be included in figure legend, not in figure title.
Figure 2: Thank you for your response. Now I understand the procedure of statistical analysis here. Again, I think the most important result in this study is the fact that 25-39 years group has a significantly higher antibody titer than 6-12 years group. However, it is only described in the text, and this Figure 2 demonstrates “negative data” (antibody titer proportions in each age group).
1) Titer distribution in each age group is recommended to be shown here, instead of the current Figure 2. This study is handling large number of samples, and therefore boxplot would be useful to illustrate antibody titer distribution in each age group. Examples can be found at (https://www.mdpi.com/2076-393X/10/9/1511), or (https://www.mdpi.com/2076-393X/10/5/734/html). The result of statistical analysis (P value or symbol) can be added to the figure.
2) Please reconsider to show the data in the current Figure 2 (titer proportions in each age group) in a table (see the comment below).
Figure 3: Explanation of local areas should be included in figure legend, not in figure title. The authors did not find statistical difference in antibody titer proportions (Chi-squared test), but found between the Northern and Southern Italy in GMT (one-way ANOVA followed by Turkey-Kramer test)? If my above understanding is correct, Figure 3 is also recommend illustrating by boxplot for titer distribution in each area.
Tables: As written in lines 161-163, several variables are investigated their association with anti-PT IgG titer. Therefore, it may be better to summarize them into one table under each variable row; Gender (male, female), Age (5 age groups), Geographical area (4 geographical areas. Each data from 13 local areas is not necessary to be shown). In addition to the 6 columns (sample number, 4 categories of antibody titer, GMT±SD), insertion of the column for P-value would be helpful to understand. Examples can be found at (https://www.mdpi.com/2076-393X/10/9/1511), or at (https://www.mdpi.com/2076-393X/10/11/1806).
Lines 237-358, Discussion: I think 23 paragraphs are too much for this research article because it is hard to follow-up the point of argument. I recommend focusing on the selected issues and discuss them based on your findings (Basically one paragraph includes one issue). However, the authors do not want to change the current construction as responded to my original review. I would therefore leave a decision to the Editor here.
Author Response
Reviewer 4
This study includes large-scale sero-epidemiology data, and the data it-self is valuable for public health. However, the manuscript is still required extensive improvements mainly in data presentation.
Thank you for the evaluation of our manuscript. We have further modified the manuscript taking into account the following specific remarks.
Specific remarks:
Lines 102-163, Materials and Methods: Some descriptions are duplicated (e.g., lines 85-87 and line134, lines 134-140 and 144-149, line 114 and 133). Please eliminate such duplicating expressions throughout the manuscript. To avoid such errors, I recommend describing methodological details under each block: e.g., 1. Study design and sample collection, 2. Measurement of antibody titers, 3. Statistical analysis.
We understand that the Reviewer refers to the clean copy as the provided numbers are not the same included in the manuscript for revisions available at the link.
We have described methodological details identifying the same as suggested and we have eliminated duplicating expressions throughout the manuscript.
In detail, now the Materials and Methods sections is as follows (pages 3-7, lines 109-246):
Materials and Methods
- Study design and sample collection
The study was designed as an in vitro, not interventional, multicenter study, promoted by the Italian Institute of Health (ISS).
The objectives of the study have been to evaluate the prevalence of antibodies to pertussis toxin (anti-PTx IgG) in the study population by age group, to evaluate the distribution of average antibody titers (GMT) (humoral immunity) by age group, to evaluate the distribution of subjects susceptible to pertussis as well as of subjects with high (>100 IU/ml) anti-PTx IgG levels by age and to evaluate the above points stratifying data accordingly to the gender and the geographical area (North, Center and South Italy). Anonymous unlinked samples of residual sera from routine laboratory testing were collected from subjects between 6 and 90 years of age without any immune-depressive condition or any acute infection or not recently undergone blood transfusion. Some demographic data of each subject, such as age, gender and geographical area of residence, were taken. Collected sera were stored at -20°C.
The total number of sera to be collected was calculated taking into account that the study has been designed to evaluate the sero-prevalence against a number of vaccine-preventable infectious diseases. The sampling protocol for each regional center was in accordance with the estimates made during the sero-epidemiological studies conducted within the European ESEN (European Sero-Epidemiological Network) project [12-16], in which samples were taken for the national sero-prevalence studies conducted in 1996, 2003-2004 and 2013- 2014. Nowadays, available samples have been collected (period June 2019 - May 2020) from 13 regional centers, (Northern Italy: Autonomous Province of Bolzano, Emilia-Romagna, Liguria, Piedmont and Veneto; Central Italy: Tuscany, Marche; Southern Italy and Islands: Basilicata, Calabria, Campania, Apulia, Sardinia, Sicily) (Figure 1).
- Measurement of antibody titers
All collected sera have been sent to the Laboratory of Hygiene of the Department of Biological and Environmental Sciences and Technologies, University of Salento, Lecce, Italy where they have been analyzed.
To determine the anti-PTx IgG levels, the classical immune-assay Serion ELISA (Institut Virion/Serion GmbH, Germany) was used. This assay permits to determine antibodies to B. pertussis specific toxin (PTx), using the first WHO International Standard Pertussis Antiserum (Human) 1st IS NIBSC code: 06/140, available since 2009 and expressing qualitative (positive/negative) and quantitative results in terms of antibody activity as international units per milliliter (IU/ml). The lower limit of detection for anti-PTx IgG is 5 IU/mL. A cut-off of 100 IU/ml is considered to be an indicator of a recent infection (occurred in the last year), while levels greater than or equal to 40 IU/ml of an infection that has occurred in recent years.
- Statistical analysis
The main statistical techniques that apply to observational studies (epidemiological statistics) have been used to analyze the data collected in the study.
In particular, the evaluation of the prevalence of anti-PTx IgG in the study population by age group have been done using the chi square test. The same test has been used to compare data stratified accordingly to geographical area (Northern, Central and Southern Italy) and region. In any case, the significance level has been set at 0.05. The analysis of the average antibody titers (GMT) by age group, gender and geographical area has been conducted using one-way ANOVA test followed by Tukey-Kramer post-hoc test. As an additional analysis, the association between the presence of anti-PT IgG antibodies and variables such as gender, age and geographical area has been assessed through a logistic regression model.
Lines 149-152: Please provide the reference for this titer categorizing.
Thank you. Please take note that references for this were already included in the Introduction section. For clarity, we have added the same references [7, 11] in page 6, line 201.
Figure 1: Explanation of color keys should be included in figure legend, not in figure title.
Thank you. We have changed Figure 1 as suggested (page 4-6).
Figure 2: Thank you for your response. Now I understand the procedure of statistical analysis here. Again, I think the most important result in this study is the fact that 25-39 years group has a significantly higher antibody titer than 6-12 years group. However, it is only described in the text, and this Figure 2 demonstrates “negative data” (antibody titer proportions in each age group).
1) Titer distribution in each age group is recommended to be shown here, instead of the current Figure 2. This study is handling large number of samples, and therefore boxplot would be useful to illustrate antibody titer distribution in each age group. Examples can be found at (https://www.mdpi.com/2076-393X/10/9/1511), or (https://www.mdpi.com/2076-393X/10/5/734/html). The result of statistical analysis (P value or symbol) can be added to the figure.
As suggested, we have replaced Figure 2 with a boxplot describing the distribution of antibody titers and the mean in the different age groups (page 9).
2) Please reconsider to show the data in the current Figure 2 (titer proportions in each age group) in a table (see the comment below).
As suggested, data relating to seroprevalence in different age groups have been included in Table 1 (page 8).
Figure 3: Explanation of local areas should be included in figure legend, not in figure title. The authors did not find statistical difference in antibody titer proportions (Chi-squared test), but found between the Northern and Southern Italy in GMT (one-way ANOVA followed by Turkey-Kramer test)? If my above understanding is correct, Figure 3 is also recommend illustrating by boxplot for titer distribution in each area.
Thank you. We have added legenda at the bottom of the figure and deleted the same from the title. Besides, as suggested, we have replaced figure 3 with a boxplot describing the distribution of antibody titers and the mean in the different geographical areas (page 11).
As described in the Material and Methods section the analysis of the average antibody titers by age group, gender and geographical area has been conducted using one-way ANOVA test followed by Tukey-Kramer post-hoc test.
Tables: As written in lines 161-163, several variables are investigated their association with anti-PT IgG titer. Therefore, it may be better to summarize them into one table under each variable row; Gender (male, female), Age (5 age groups), Geographical area (4 geographical areas. Each data from 13 local areas is not necessary to be shown). In addition to the 6 columns (sample number, 4 categories of antibody titer, GMT±SD), insertion of the column for P-value would be helpful to understand. Examples can be found at (https://www.mdpi.com/2076-393X/10/9/1511), or at (https://www.mdpi.com/2076-393X/10/11/1806).
As suggested, data relating to seroprevalence by age group and geographical area have been included in Table 1 together with those relating to gender. A sixth column has also been added showing the p-value calculated using the chi-squared test (page 8).
Lines 237-358, Discussion: I think 23 paragraphs are too much for this research article because it is hard to follow-up the point of argument. I recommend focusing on the selected issues and discuss them based on your findings (Basically one paragraph includes one issue). However, the authors do not want to change the current construction as responded to my original review. I would therefore leave a decision to the Editor here.
As we have already pointed out in the rebuttal letter enclosed with the previous revision of our manuscript, some changes were included in the Discussion section taking into account the points raised also by another Reviewer, who asked to add some more details and new parts. Besides, some findings have been discussed in more detail in order to clarify the results of our study. For these reasons, we would not like to change the structure of the Discussion section.